# On the Feasibility of Compressing Certifiably Robust Neural Networks

**Pratik Vaishnavi, Veena Krish**
Stony Brook University
{pvaishnavi,kveena}@cs.stonybrook.edu

**Farhan Ahmed, Kevin Eykholt**
IBM Research
{farhan.ahmed,kheykholt}@ibm.com

**Amir Rahmati**
Stony Brook University
amir@cs.stonybrook.edu

## Abstract

Knowledge distillation is a popular approach to compress high-performance neural networks for use in resource-constrained environments. However, the threat of adversarial machine learning poses the question: Is it possible to compress adversarially robust networks and achieve similar or better adversarial robustness as the original network? In this paper, we explore this question with respect to *certifiable robustness defenses*, in which the defense establishes a formal robustness guarantee irrespective of the adversarial attack methodology. We present our preliminary findings answering two main questions: 1) Is the traditional knowledge distillation sufficient to compress certifiably robust neural networks? and 2) What aspects of the transfer process can we modify to improve the compression effectiveness? Our work represents the first study of the interaction between machine learning model compression and certifiable robustness.

## 1 Introduction

The existence of adversarial inputs [8, 20], *i.e.,* imperceptibly perturbed inputs that reliably cause erroneous outputs, have heightened concerns regarding the use of neural networks in sensitive real-world settings. In response, several methods have been proposed to harden neural networks against such inputs, enhancing their reliability in adversarial environments [15, 13, 19, 5]. One such category of methods focuses on computing the size of the largest neighborhood around a given input within which a classifier's output remains constant. Such a classifier is said to be *certifiably robust* for the given input inside this neighborhood. Since these methods provide a guarantee of robustness, they are highly desirable in safety- and privacy-critical applications such as payment systems, access control systems, self-driving cars, and security surveillance [14]. However, the expanded usage of neural networks in resource-limited computing platforms, such as IoT systems, has introduced a new challenge in this space: when training to generalize over adversarial data rather than the standard data distribution, larger networks are necessary [4, 16], making robust networks more resource demanding compared to their non-robust counterparts.

Knowledge distillation (KD) [3, 9] is a technique which uses a teacher-student training pipeline to compress the performance of larger networks into a smaller architecture. On the standard (*i.e.,* non-adversarial) classification task, KD improves the performance of the smaller student network and often results in performance compared to the large teacher network. Thus, a question arises: can knowledge distillation be used in an adversarial context to compress highly robust neural networks? Several works have explored this question with respect to *empirical robustness*, in which adversarial

2022 Trustworthy and Socially Responsible Machine Learning (TSRML 2022) co-located with NeurIPS 2022.

robustness is measured with respect to a specific attack algorithm [7, 1, 22, 23]. We note, though, no work has explored this question with respect to certifiable robustness, which is more given that it establishes a security guarantee irrespective of the attack methodology. To address the gap in literature, we perform the first study on compressing certifiably robust networks.

In this paper, we make the following **contributions**:

- We study the various strategies used for KD in context of certifiable robustness. We discover shortfalls in naive application of KD in these settings.
- We propose a more effective strategy for distilling certified robustness into small networks, which allows us to bridge the gap between the robustness of the student and the teacher.
- We identify that distilling certifiable robustness imposes stricter requirements than distilling standard performance or empirical robustness. Specifically, a smaller size gap between the student and the teacher networks might be required for effective distillation.

## 2 Background and Related Work

In this work, we study the certifiable robustness of neural network based image classifiers trained using knowledge distillation (KD). We use this section to define our problem, provide relevant background, and discuss related prior works.

### 2.1 Preliminaries

Consider a neural network classifier $f$ parameterized by $\theta$ that is trained to map a given image $x \in \mathbb{R}^d$ to a set of discrete labels $\mathcal{Y}$ using a set of *i.i.d.* samples $\mathcal{S} = \{(x_1, y_1), (x_1, y_1), \cdots, (x_n, y_n)\}$ drawn from some data distribution. The output of the classifier can be written as $f(x; \theta) = \mathrm{argmax}_{c \in \mathcal{Y}} z_c(x; \theta)$. Here $z(x; \theta)$ is the softmax output of the classifier and $z_c(x; \theta)$ denotes the probability that image $x$ belongs to class $c$. The process of training of a classifier involves minimizing the cross-entropy loss on the standard data distribution, which is approximated through $\mathcal{S}$.

### 2.2 Adversarial Robustness

In the $\ell_p$-norm space, an adversarial perturbation is defined as any perturbation $\delta \in \mathbb{R}^d$ such that $\|\delta\|_p \leq \epsilon$, that an adversary can use to change the classifier's output *i.e.,* $f(x; \theta) \neq f(x + \delta; \theta)$. Here, $\epsilon$ defines the adversarial perturbation budget in terms of the maximum allowed magnitude of the perturbation vector.

The adversarial robustness of a neural network is often characterized by its *empirical* or *certified* robustness. The empirical adversarial robustness of a neural network is the accuracy of the network against adversarial samples generated by a given attack algorithm. Many early proposed defenses that were thought to be effective based on their reported empirical robustness were later found to have poor true robustness [2]. In contrast, the certified adversarial robustness of a neural network represents the true worst-case performance and is measured by the lower bound accuracy of the network against *all* adversarial samples within a given $\epsilon$ neighborhood.

#### 2.2.1 Certified Robustness

The certified robustness of a network can be measured by its lower bound accuracy within a pre-determined neighborhood. Alternatively, it can be measured based on the radius of the largest neighborhood within which a classifier's output remains (correct and) constant. For a given input $x$, the classifier $f$ is said to be certifiably robust if $f(x; \theta)$ is provably constant within some large neighborhood around $x$. The radius of this neighborhood (or, robust radius) is defined using the $\ell_p$-norm metric as follows:

$$R(f; x, y) = \begin{cases} \inf_{f(x'; \theta) \neq f(x; \theta)} \|x' - x\|_p & \text{, when } f(x; \theta) = y \\ 0 & \text{, when } f(x; \theta) \neq y \end{cases} \tag{1}$$

Intuitively, the robust radius $R(f; x, y)$ establishes a region within which the classifier's prediction remains constant, ensuring that an adversary with a budget $\epsilon \leq R$ can not succeed. Therefore, training networks to maximize the robust radius would harden a classifier against manipulations of all sorts, including adversarial ones. However, as computing the robust radius of a neural network for a given input $x$ is NP-hard [12], recent certified training methods instead propose computing a lower bound of the robust radius, known as the *certified radius*.

***Randomized Smoothing.*** Cohen *et al.* [6] presented a scalable method for creating certifiably robust image classifiers using randomized smoothing. This involves converting a given classifier (termed *base classifier*) into a *smooth classifier*.

**Definition 2.1.** For a given (base) classifier $f$ and $\sigma > 0$, the corresponding smooth classifier $g$ is defined as follows:

$$g(x;\theta) = \underset{c \in \mathcal{Y}}{\operatorname{argmax}} P_{\eta \sim \mathcal{N}(0,\sigma^2 I)}(f(x+\eta;\theta) = c) \tag{2}$$

Simply put, $g$ returns the class $c$, which has the highest probability mass under the Gaussian distribution $\mathcal{N}(x,\sigma^2 I)$. The authors provide theoretical proof for the certified robustness of a smooth classifier. This theoretical work can be summarized using Theorem 2.2.

**Theorem 2.2.** *Let $f : \mathbb{R}^d \mapsto \mathcal{Y}$ be a classifier and $g$ be its smoothed version (as defined in Equation 2). For a given input $x \in \mathbb{R}^d$ and corresponding ground truth output $y \in \mathcal{Y}$, if $g$ correctly classifies $x$ as $y$ such that*

$$P_\eta(f(x+\eta;\theta) = y) \geq \max_{y' \neq y} P_\eta(f(x+\eta;\theta) = y') \tag{3}$$

*then $g$ is provably robust at $x$ within the certified radius $R$ given by:*

$$CR(g;x,y) = \frac{\sigma}{2}[\Phi^{-1}(P_\eta(f(x+\eta;\theta) = y)) - \Phi^{-1}(\max_{y' \neq y} P_\eta(f(x+\eta;\theta) = y'))] \tag{4}$$

*where, $\Phi$ is the c.d.f. of the standard Gaussian distribution.*

Furthermore, the authors demonstrate that training the base classifier to minimize cross-entropy loss on inputs perturbed with Gaussian noise (Gaussian data augmentation) increases the certified radius of the smooth classifier. More generally, improving the base classifier's robustness to Gaussian noise is a successful strategy for increasing certified radius and has been utilized by several other prior works [18, 21, 11].

## 2.3 Knowledge Distillation (KD)

For the standard classification task, small neural networks are able to learn similar classification functions as large ones through a process known as knowledge distillation [3, 9]. Traditional KD involves training the small network (*student*) to mimic the outputs of a much larger network (*teacher*) for a given task (*e.g.,* classification). The student's training objective, also referred to as the *distillation objective*, is formalized as follows:

$$\min_\theta \mathbb{E}_{(x,y) \sim \mathcal{S}}[(1-\alpha)\mathcal{L}_{CE}(z(x;\theta),y) + \alpha t^2 \mathcal{L}_M(z^t(x;\theta), z^t(x;\phi))] \tag{5}$$

where $z(x;\theta)$ and $z(x;\phi)$ are the softmax outputs of the student $S$ and teacher $T$, respectively; $\mathcal{L}_{CE}$ is the cross-entropy loss; $\mathcal{L}_M$ is the "mimic loss" (KL-Divergence, Euclidean distance *etc*); $\alpha$ is an hyperparameter used to weigh the two loss terms; and $t$ is the softmax temperature. The value of $\alpha$ is usually set to 1, implying that the student is solely trained to mimic the teacher and in the process learns to perform well on the task. Training with the supervision of a teacher improves the student's performance as compared to training it independently. This is due to the small student benefits from the inter-class relationships learned by the large teacher (with higher modelling capacity). By distilling the knowledge of a large network into a small network, we essentially perform model compression, as the small network encodes the performance of the large network but with fewer parameters. This, in turn, allows for the use of high performance networks in resource-restricted devices.

## 2.4 Adversarial Robustness and Knowledge Distillation

Using the lessons learned from KD, several successful attempts have been made at improving the adversarial robustness of small networks by training them under the supervision of larger, robust networks. Goldblum *et al.* [7] propose *Adversarially Robust Distillation* (ARD), that combines KD with Adversarial Training [15]: the student is trained to match the teacher's outputs on adversarial inputs. ARD improves the robustness of small networks against gradient-based attacks relative to standalone adversarial training. Zi *et al.* [23] propose *Robust Soft Label Adversarial Distillation* (RSLAD), which improves upon ARD by using soft labels from a robust teacher rather than hard labels in all supervision loss terms. An important commonality between all these prior works is that they limit their experimentation to adversarial training, which is an empirical robustness method. Therefore, the robustness claims made by them can potentially be invalidated by a future adversary.

**Table 1:** The mimic loss used in different distillation objectives proposed by prior works. The function $z$ represents softmax outputs of the student and teacher network respectively, and $t$ represents the temperature parameter. Note that since $\alpha$ is usually set to 1 (see Equation 5), we only report the loss terms that the student is actually trained with.

| METHOD | $\mathcal{L}_{\mathcal{M}}$ |
|---|---|
| KD [3, 9] | KL-DIV$(z^t(x;\theta),z^t(x;\phi))$   OR   $\|z(x;\theta)-z(x;\phi)\|_2$ |
| ARD [7]* | KL-DIV$(z^t(x+\delta;\theta),z^t(x;\phi))$ |
| RSLAD [23]* | KL-DIV$(z(x;\theta),z(x;\phi))+$KL-DIV$(z(x+\delta;\theta),z(x;\phi))$ |

*For $\delta$ we use Gaussian noise instead of adversarial noise.

## 3  Distilling Certified Robustness

In order to promote deployment of safe machine learning models in resource-limited settings, it is important to study if certifiably robust neural networks can be effectively compressed. Knowledge distillation (KD) is one of the most effective approaches for doing this. Therefore, in this section we examine the effectiveness of KD towards compressing certifiably robust neural networks. We begin by describing our experimental setup in Section 3.1. In Section 3.2, we study the effectiveness of existing distillation objectives in distilling certified robustness. In Section 3.3, we propose a distillation objective using the traditional KD strategy that improves the distillation process by addressing the shortcoming of existing objectives.

### 3.1  Experimental Setup

In our experiments, we focus on certified robustness of image classifiers in the $\ell_2$-space. Following the work by Cohen *et al.* [6], we use randomized smoothing to achieve certifiably robust classifiers. All our experiments are conducted using the CIFAR-10 dataset. To measure certified robustness, we follow prior works and report certified accuracy (or, the prediction accuracy of the smooth classifier) at different $\ell_2$ radius [6, 18, 11]. The certified accuracy at radius$= 0$ is equivalent to the prediction accuracy of the smooth classifier on clean inputs. Additionally, we report the *average certified radius* (ACR) computed over the entire test set [21]. Our code is implemented in PyTorch [17] and is publicly available at `https://github.com/Ethos-lab/crd-tsrml22`. [1]

### 3.2  Can existing distillation objectives be used to distill certified robustness?

Knowledge distillation is effectively a method for "transferring" the knowledge of one network to another. Traditionally, this is achieved by training one network to mimic the other using some sort of mimic loss (*i.e.,* $\mathcal{L}_{\mathcal{M}}$ from Equation 5). Prior works on distilling adversarial robustness [7, 23, 22] also follow the traditional KD strategy, but propose different versions of $\mathcal{L}_{\mathcal{M}}$ (see Table 1). In this section, we evaluate whether these existing distillation objectives can be used to distill certified robustness. Note that since ARD and RSLAD were designed for adversarial training [15], they require computing adversarial noise ($\delta$) using first-order attacks like PGD [15]. This makes them incompatible with the certified robustness methods that we wish to study. To make these methods usable for our study, we use Gaussian noise in place of the adversarial noise term ($\delta$) in the respective distillation objectives.

For experimentation, we use a ResNet-110 network as the teacher and train it on the CIFAR-10 dataset. To obtain non-trivial certified robustness for this network, we train it using the Gaussian data augmentation method proposed by Cohen *et al.* [6]. We then distill its robustness to a much smaller ResNet-20 network using the different existing distillation objectives. The results are summarized in Table 2. For comparison, we also report the student and teacher network's robustness when trained independently using Gaussian data augmentation. The first observation we make is that traditional distillation (KD) completely fails at distilling certified robustness as the student exhibits trivial ACR and certified accuracy for all values of $r$. The student network trained with RSLAD appears to have non-trivial certified robustness, however, the distilled ResNet-20 network exhibits poorer robustness than the one trained independently. This implies that the RSLAD objective is also unsuitable for our use case. Only the ARD objective seems to be successful at distilling certified robustness as the distilled ResNet-20 exhibits higher robustness than a ResNet-20 trained independently.

---

[1]For certification, we borrow code from Cohen *et al.* [6]: `https://github.com/locuslab/smoothing`

**Table 2:** Comparing the robustness of a student network trained using different variants of knowledge distillation. We denote the distillation process as "teacher $\xrightarrow{\text{method}}$ student". Among all the methods, only ARD is able to successfully distill certified robustness, presenting higher robustness than the independently-trained student.

|  | ACR | 0.00 | 0.25 | 0.50 | 0.75 |
|---|---|---|---|---|---|
| RESNET-110 | 0.486 | 81.41 | 67.75 | 49.67 | 32.37 |
| RESNET-20 | 0.451 | 79.62 | 63.78 | 45.65 | 28.01 |
| RESNET-110 $\xrightarrow{\text{KD}}$ RESNET-20 | 0.090 | 10.93 | 10.16 | 9.86 | 9.03 |
| RESNET-110 $\xrightarrow{\text{RSLAD}}$ RESNET-20 | 0.431 | 77.46 | 61.98 | 43.57 | 25.62 |
| RESNET-110 $\xrightarrow{\text{ARD}}$ RESNET-20 | 0.456 | 76.50 | 62.80 | 46.87 | 30.29 |

**Table 3:** Evaluating the effectiveness of CRD in distilling certified robustness. CRD performs better than distillation objectives proposed by prior works. Furthermore, CRD trained student exhibits comparable robustness to its teacher (Table 2, $1^{st}$ row).

|  | ACR | 0.00 | 0.25 | 0.50 | 0.75 |
|---|---|---|---|---|---|
| RESNET-110 $\xrightarrow{\text{CRD}}$ RESNET-20 | 0.483 | 80.06 | 66.19 | 49.62 | 32.74 |

## 3.3 Adapting traditional KD strategy to distill certified robustness

From the results in Table 2, we observe that even for the best performing objective (*i.e.,* ARD), there exists a gap between the teacher's and the student's robustness. In this section we explore whether it is possible to bridge this gap while using the traditional KD strategy of mimicking outputs. We note that formulation of $\mathcal{L}_M$ used by prior works on distilling adversarial robustness was motivated by wanting the student to learn a similar output distribution for clean and adversarial inputs (generated using some attack). This motivation, however, doesn't translate to our use case very well. In the randomized smoothing paradigm, higher certified robustness comes from higher robustness to Gaussian noise. In fact, Jeong *et al.* [10] note that there is a direct correlation between the robustness of a smooth classifier and its prediction confidence (tied to confidence of base classifier on inputs perturbed with Gaussian noise). Based on this, we propose the following $\mathcal{L}_M$ which is tailored for certified robustness distillation:

$$\min_{\theta} \mathbb{E}_{(x,y)\sim\mathcal{S};\delta\sim\mathcal{N}(0,\sigma^2 I)}[(1-\alpha)\mathcal{L}_{CE}(z(x+\delta;\theta),y) + \alpha t^2 \|z^t(x+\delta;\theta) - z^t(x+\delta;\phi)\|_2] \quad (6)$$

We refer to this distillation objective as *Certified Robust Distillation* (CRD). Simply put, we are training the student to mimic the teacher's output not only at the given input $x$, but also in the Gaussian neighborhood around it. The robustness of a ResNet-20 distilled from a ResNet-110 using CRD is reported in Table 3. Comparing with the results in Table 2, we observe that CRD is successful at bridging the gap between the robustness of the student and the teacher. This makes CRD a more successful objective for distilling certified robustness than objectives proposed by prior works.

## 4 Limitations of CRD

In this section, we address the existing limitations of CRD. It is known that robustness training requires the network to learn more complicated functions than standard training [4, 16]. Therefore, we suspect that it might be more difficult to distill networks with high robustness as compared to distilling networks with high standard performance (*e.g.,* accuracy on test set). To investigate this, we repeat the experiment from previous section, this time training the teacher to be more robust by using training methods that are better than Gaussian data augmentation training. Specifically, we use MACER [21] and SmoothMix [10]. The results are reported in Table 4. For comparison, we also report the robustness of student and teacher networks independently trained using MACER and SmoothMix. Overall, we observe that for all teacher training methods that we use in this section, the process of distillation is not as effective as it was in the previous section. In all cases, there is a large gap between the robustness of the student and the teacher networks. Furthermore, only in one case (*i.e.,* MACER) we observe that the distilled ResNet-20 has higher robustness than the independently trained ResNet-20.

**Table 4:** Evaluating the effectiveness of CRD in distilling certified robustness from ResNet-110 teachers with progressively higher robustness. It is harder for a student to mimic a more robust teacher.

| | ACR | 0.00 | 0.25 | 0.50 | 0.75 |
|---|---|---|---|---|---|
| MACER [21] | | | | | |
| RESNET-110 | 0.531 | 79.11 | 68.39 | 55.90 | 40.61 |
| RESNET-20 | 0.507 | 76.44 | 65.81 | 52.87 | 38.75 |
| RESNET-110 $\xrightarrow{\text{CRD}}$ RESNET-20 | 0.508 | 78.30 | 66.80 | 53.15 | 37.75 |
| SMOOTHMIX [10] | | | | | |
| RESNET-110 | 0.550 | 76.89 | 68.25 | 57.42 | 46.26 |
| RESNET-20 | 0.522 | 75.55 | 65.53 | 54.72 | 42.62 |
| RESNET-110 $\xrightarrow{\text{CRD}}$ RESNET-20 | 0.514 | 76.33 | 65.85 | 53.83 | 40.28 |

**Table 5:** Testing the network capacity limitation of CRD using SmoothMix [10]. Networks of larger sizes are required to effectively mimic teachers possessing high certified robustness.

| | ACR | 0.00 | 0.25 | 0.50 | 0.75 |
|---|---|---|---|---|---|
| RESNET-110 | 0.550 | 76.89 | 68.25 | 57.42 | 46.26 |
| RESNET-32 | 0.537 | 76.44 | 67.17 | 56.19 | 44.00 |
| RESNET-110 $\xrightarrow{\text{CRD}}$ RESNET-32 | 0.530 | 76.81 | 67.57 | 55.52 | 42.48 |
| RESNET-44 | 0.545 | 76.55 | 67.33 | 57.18 | 45.85 |
| RESNET-110 $\xrightarrow{\text{CRD}}$ RESNET-44 | 0.541 | 77.34 | 68.11 | 56.84 | 43.92 |
| RESNET-56 | 0.545 | 77.01 | 68.17 | 56.89 | 45.05 |
| RESNET-110 $\xrightarrow{\text{CRD}}$ RESNET-56 | 0.547 | 77.60 | 68.24 | 57.72 | 44.97 |

We run additional experiments to further investigate this student network capacity limitation of CRD. Starting with a ResNet-110 teacher trained using SmoothMix, we distill it into networks of various sizes using CRD. The results for this experiment are reported in Table 5. We observe that as we increase the size of the student network, the effectiveness of the distillation process improves. For ResNet-56, which is about half the size of ResNet-110, we observe that CRD succeeds at achieving comparable robustness between the student and the teacher. The gap between the robustness of the student and the teacher gets progressively worse as we reduce the size of the student. These results further corroborate that CRD, in its current form, is unable to distill complicated functions learnt by state-of-the-art robustness methods into networks of certain size. This result in unlike what prior works have reported in context of distilling both standard performance and adversarial robustness where we see that distillation can be successfully performed between student and teacher networks with larger size differences than the ones we use (*e.g.,* WideResNet-34-10 to ResNet-20) [3, 7, 23].

## 5 Conclusion & Future Work

In this paper, we presented the first study of knowledge distillation (KD) in the context of certified robustness. Specifically, we focus on randomized smoothing based (probabilistic) certified robustness. We tested different existing distillation objectives that were designed to distill standard performance or (empirical) adversarial robustness in terms of how effective they are in distilling certified robustness. Based on these results, we proposed a distillation objective (CRD) tailored for distilling certified robustness. However, CRD suffers from a network capacity limitation which makes it impractical to use with state-of-the-art certified training methods; further research is needed to address this shortcoming. We believe our preliminary investigation will serve as a useful starting point for future works on compressing certifiably robust machine learning models.

## Acknowledgement

This work was supported by the Office of Naval Research under grants N00014-20-1-2858 and N00014-22-1-2001, Air Force Research Lab under grant FA9550-22-1-0029, and NVIDIA 2018 GPU Grant. Any opinions, findings, or conclusions expressed in this material are those of the authors and do not necessarily reflect the views of the sponsors.

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

## A    Additional Results

To provide a comprehensive understanding of CRD, we further dissect the training objective and study the effect of changing its different components on the effectiveness of the distillation process. First, we look at the loss function used to enforce the matching of the student's output to the teacher. In all of our experiments so far, we do this matching using $\ell_2$ distance on the pre-softmax outputs (logits) [3]. However, another popularly used strategy involves using KL-Divergence on the softmax outputs [9]. Therefore, we compare the effectiveness of CRD when using KL-Divergence to its $\ell_2$ loss variant. Furthermore, we also compare with a variant using cosine distance in the logit space.

The overall loss function (see Equation 6) requires two hyperparameters: (1) $\alpha$: which controls the contribution of the standard cross-entropy loss and the "mimic loss" towards the total loss, and (2) temperature $t$: the logits are divided by this value before feeding them to the softmax function. To perform a thorough comparison, we train several classifiers using different values of these hyperparameters. The results are summarized in Table 6.

Overall, we do not notice any significant improvements on matching logits compared to matching softmax outputs. In fact, $\ell_2$ loss on logits yields comparable robustness and prediction accuracy on clean inputs relative to KL-Divergence loss on softmax outputs. Specifically, we observe that not using cross-entropy loss during training ($\alpha = 1$) does not have any adverse effect on the student's overall performance. The students trained using cross-entropy loss ($\alpha = 0.9$) exhibit similar levels of prediction accuracy on clean inputs but reduced robustness. When using the KL-Divergence loss, we observe that using higher value of temperature $t$ yields higher robustness. However, this improvement plateaus quickly, at around $t = 16$.

**Table 6:** Ablation results for hyperparameter usage with the CRD loss function. Here, we distill a ResNet-110 classifer into a ResNet-20 classifier on the CIFAR-10 dataset.

| Loss | $\alpha$ | t | ACR | Radius | | | |
|---|---|---|---|---|---|---|---|
| | | | | 0.00 | 0.25 | 0.50 | 0.75 |
| KL-Divergence | | 1.0 | 0.461 | 80.15 | 65.09 | 47.01 | 28.67 |
| | 1.0 | 4.0 | 0.480 | 80.33 | 66.20 | 49.40 | 32.01 |
| | | 16.0 | 0.481 | 79.83 | 65.95 | 49.47 | 32.52 |
| | | 1.0 | 0.462 | 80.01 | 65.17 | 47.15 | 29.05 |
| | 0.9 | 4.0 | 0.480 | 80.30 | 66.44 | 49.26 | 32.03 |
| | | 16.0 | 0.480 | 79.87 | 66.00 | 49.24 | 32.47 |
| $\ell_2$ distance | 1.0 | 1.0 | 0.482 | 79.57 | 65.99 | 49.57 | 32.79 |
| | 0.9 | 1.0 | 0.481 | 79.74 | 65.93 | 49.39 | 32.70 |
| Cosine distance | 1.0 | 1.0 | 0.470 | 79.34 | 65.17 | 48.32 | 31.19 |
| | 0.9 | 1.0 | 0.469 | 79.89 | 65.58 | 47.79 | 30.46 |

