# OpenReview forum: "On the Feasibility of Compressing Certifiably Robust Neural Networks"
_NeurIPS.cc/2022/Workshop/TSRML — TSRML2022_

### Official Review · Reviewer_iiyA · 2022-10-11
**Interesting problem; improvement needed**

**Overall Rating:** 6

**Summary:**

The paper studies the knowledge distillation approach for transfer certifiable robustness. The paper proposes a CRD training loss to improve distillation. CRD is effective for teacher models trained with Gaussian noise but is less effective for teacher models trained with more sophisticated and advanced algorithms.

**Strengths:**

1. First to study the knowledge distillation problem in the context of certifiable robustness
2. The proposed CRD is simple and effective for teacher models trained with Gaussian noises.


**Weaknesses:**

1. the current version of CRD does not work well for more sophisticated teacher models trained with more advanced smoothing techniques. There is still a gap to close for this idea to be meaningful and practical for the community.
2. Suggestion: how about other certifiably robust techniques (other than randomized smoothing)?  For example, tools like network verification? https://github.com/Verified-Intelligence/alpha-beta-CROWN
3. Suggestion: could you also report the clean accuracy (model accuracy on benign images) to understand how distillation affects the model utility?
4. Abused notation: z^t. the subscript is used for temperature (Table 1) and class label (Section 2.1)


**Overall Recommendation:**

The paper raised an interesting research question and made a successful preliminary attempt (CRD). Though CRD does not work for more robust teachers at this moment, I think there is a good chance to improve CRD in future work.

**Review Confidence:**

4: The reviewer is confident but not absolutely certain that the evaluation is correct

---

### Official Review · Reviewer_fZXH · 2022-10-21
**Practical improvement**

**Overall Rating:** 6

**Summary:**

This paper discusses the use of knowledge distillation to compress networks while maintaining good certified robustness. Existing knowledge distillation methods are investigated and a modification is proposed to improve the ACR. Limitations are discussed which concludes the work.

**Strengths:**

The paper discussed an interesting question and gave useful results. This work has the first study on compressing certifiably robust networks and use experiments to demonstrate the effectiveness of the proposed improvements. The limitation discussion is also helpful.

**Weaknesses:**

1. Current work only focuses on certified robustness of image classifiers in the $\ell_2$ space and
2. only conducts experiments on CIFAR10.
3. It is not clear why ARD and RSLAD are not compatible with the certified robustness methods.
4. It would be helpful if the authors can explain the choice of $\alpha$ in Eqn 6 and how it affects the results.
5. Please also consider to report the standard deviation of the experiment results.

**Overall Recommendation:**

Marginally above acceptance threshold. Despite the room for improvements, this paper is still interesting and useful for the community.

**Review Confidence:**

3: The reviewer is fairly confident that the evaluation is correct

---

### Official Review · Reviewer_TzAL · 2022-10-22
**Successful compression of certified robustness neural networks using knowledge distillation**

**Overall Rating:** 7

**Summary:**

This paper focuses on compression of certified robust networks, where it uses randomized smoothing with knowledge distillation to achieve certified robustness. It show that canonical distillation techniques achieve suboptimal performance, and propose a new distillation techniques to mitigate the challenge.

**Strengths:**

- This paper is written well and very easy to erase.
- It significantly outperforms pervious works.
- Even more it brings student network performance very close to the larger teacher networks.

**Weaknesses:**

- The emphasis on whether knowledge distillation in certified robust training is a uniquely challenging problem is also missing. E.g., wouldn’t the similar knowledge distillation technique would be effective for empirical robustness with adversarial training [1] or certified robustness with bound-propagation networks [2, 3]. Expanding beyond the current randomized smoothing approach would increase the impact of the paper.
- The current method also suffers from diminishing returns with larger models. While it outperforms other techniques with smaller ResNets, its benefit goes down as the size of ResNets increases.
- Aforementioned demising returns limits the capability of proposed method to smaller student models. E.g., would the method improve performance of state of the art WideResNet-28-10 if we distill knowledge from larger WideResNet—70-16 models? Any benefit of proposed approach on larger models would further increase its impact

1. Madry, Aleksander, Aleksandar Makelov, Ludwig Schmidt, Dimitris Tsipras, and Adrian Vladu. "Towards deep learning models resistant to adversarial attacks." arXiv preprint arXiv:1706.06083 (2017).
2. Weng, Lily, Huan Zhang, Hongge Chen, Zhao Song, Cho-Jui Hsieh, Luca Daniel, Duane Boning, and Inderjit Dhillon. "Towards fast computation of certified robustness for relu networks." In International Conference on Machine Learning, pp. 5276-5285. PMLR, 2018.
3. Zhang, Huan, Hongge Chen, Chaowei Xiao, Sven Gowal, Robert Stanforth, Bo Li, Duane Boning, and Cho-Jui Hsieh. "Towards stable and efficient training of verifiably robust neural networks." arXiv preprint arXiv:1906.06316 (2019).


**Overall Recommendation:**

I recommend for acceptance since the paper proposes a novel approach towards knowledge distillation. However, there remains a large room for improvement in full version of the paper.

**Review Confidence:**

5: The reviewer is absolutely certain that the evaluation is correct and very familiar with the relevant literature

---

### Decision · Program_Chairs · 2022-10-23

Accept